# Does perceived quality mediate the relationship between country of origin image and consumer buying behaviour in Nigerian hypermarkets?

José Moleiro Martins[1,2], Shedrack Chinwuba Moguluwa[3]*, João Luis Lucas[4], Chiemelie Benneth Iloka[5], Mário Nuno Mata[1,2]

1 ISCAL, Instituto Superior de Contabilidade e Administração de Lisboa, Instituto Politécnico de Lisboa, Lisbon, Portugal, 2 Business Research Unit (BRU-IUL), Instituto Universitário de Lisboa (ISCTEIUL), Lisbon, Portugal, 3 Department of Marketing, Faculty of Business Administration, University of Nigeria Enugu Campus (UNEC), Nsukka, Nigeria, 4 Departamento de Gestão, Universidade de Évora, Évora, Portugal, 5 Department of Marketing, Faculty of Management Sciences, Enugu State University of Science and Technology (ESUT), Enugu, Nigeria

* shedrack.moguluwa@unn.edu.ng

**Data Availability Statement:** All relevant data can be found here: https://doi.org/10.34740/KAGGLE/DSV/4972216.

## Abstract

The aim of this study was to examine the relationship between the country of origin image of a product and consumers' purchasing behavior in Nigerian hypermarkets, as well as the mediating role of perceived price in this relationship. The research considers four aspects of purchasing behavior: information search, product evaluation, purchase decision-making, and post-purchase behavior. A total of 1272 participants completed an online survey, and the data collected was analyzed using structural equation modeling. The results indicated that the country of origin image of a product has a significant and positive impact on consumers' information search, product evaluation, purchase decision-making, and post-purchase behavior. The study also found that perceived quality does not mediate the impact of country of origin image on information search and product evaluation, but it does play a significant and positive role in purchase decision-making and post-purchase behavior. As a result, it is important for Nigerian hypermarkets to be aware of the origin of the products they import, as importing from countries with a positive image can enhance their performance. Furthermore, perceived quality can mediate purchase and post-purchase decisions.

## 1. Introduction

For several decades, research studies examining the effect of Country-of-Origin (COO) on product perception have been a critical aspect of international marketing research. There is a substantial body of evidence that supports the notion that COO has a significant effect on consumer perception, which ultimately impacts their purchasing behavior [1, 2]. However, the magnitude and direction of this effect may vary across different product categories and are influenced by moderators and antecedents of the COO concept, including consumer product

**Funding:** This research was supported by Instituto Politécnico de Lisboa. The funders had no role in study design, data collection and analysis, decision to publish, or preparation of the manuscript.

and country knowledge, and their level of involvement in purchasing decisions [3–6]. All these factors influence consumers' attitudes towards a country, which can, in turn, influence their perception of the product's country of origin [7].

Although previous studies have primarily focused on investigating country-of-origin image as a homogeneous national construct, recent research has shown that regional differences and tendencies exist in a country's image, which can significantly influence how people perceive products from that country [8, 9]. According to Šapić et al. [10], the concept of "country of origin image" is one of the most studied concepts by social scientists, which emerged from manufacturers' decision to internationalize their business operations and offer their products in foreign markets.

Consumers' use of information about the product's country of origin is heavily influenced by their previous experience with the product or any other product originating from that country, the type of product in question, and positive or negative stereotypes about specific countries [11].

With the emergence of globalization and the internet, the economic environment has significantly transformed, leading to a considerable growth in the number of brands available to consumers. As a result, people associate themselves with images of companies, nations, and goods that affect their perception and purchasing behavior [12]. COO is identified as the "made-in" image, representing the nation where the product was manufactured or assembled. Consumers' tendency to stereotype nations can influence this perception, whereby products from certain countries are viewed positively or negatively, such as products from France viewed as luxury, while products from underdeveloped countries are viewed negatively [13].

Customers are more likely to repurchase products from countries they have a favorable opinion of, due to their perception of the products' quality, craftsmanship, and technological advancement. Empirical research studies by Ahmed & d'Astous [3], Coudounaris [14], and Karimov & El-Murad [15] support this finding.

According to Business Day Intelligence [16], Nigeria's organized retail sector has grown considerably in the past two decades, generating 16% of the country's GDP in 2006. With notable players entering the Nigerian market, such as Shoprite, Game Stores, Next, and Spar, the country's hypermarket industry is rapidly expanding. Mwamba and Qutieshat [17] note that the primary attraction for multinational hypermarkets is Nigeria's expected population of 400 million by 2050, making it the world's third-most populated nation.

Past research on the impact of country of origin image has limitations, such as being mainly conducted in developed nations and being focused on specific product categories. Limited research has been done on this topic specifically in Nigeria and none on hypermarkets. To address these gaps, this study aims to explore the significance of country of origin image on consumer behavior in Nigerian hypermarkets across all product categories, and examine whether this relationship is mediated by perceived quality. The study intends to contribute to a better understanding of regional differences in country of origin image and their respective economies. Past research by Sharma [18], and Tseng and Balabanis [19] support the need for more comprehensive studies on this topic.

## 2. Literature review and hypothesis development

### 2.1 Country of origin (made-in)

Nagashima [20] is a forerunner in the study of the influence of country of origin and defines it as the image, reputation, and stereotype that consumers, both individual and corporate, associate with products from a specific country. This image is established by factors such as national characteristics, product representativeness, history, tradition, and the economic and political

background of the country [20]. It represents a group of distinct, informational, and descriptive convictions that consumers hold about a specific country and impact their purchasing choices for products from that country [21]. The country of origin image has a similar influence on consumers' assessment of products as other factors [22].

Country of origin is a complex category that influences consumers and leads them to establish an impression of products that they know are associated with a particular country. This impression is then applied to other products from the same country [23]. The term "country of origin" pertains to the place where a product is produced or assembled [24]. In today's global business environment, many products have components made in various countries, but they are assembled in a particular country. Manufacturers ensure that the final country of assembly, regarded as the country of origin, is globally viewed as being of excellent quality, trustworthy, and technologically advanced. By doing so, they can leverage the image of the country of origin and its influence on consumer perception and decision-making when purchasing foreign products.

The image of country of origin is frequently derived from the consumer's direct experience with the country or information obtained from others [25]. Nevertheless, the effect of country of origin on consumer behavior differs from one country to another due to variations in economic, socio-cultural, and other factors.

## 2.2 Characteristics of country of origin image

Numerous studies [3, 26–28] have examined the concept of country image and the impact of country of origin from various perspectives. The term "image" generally refers to the personality or identity that people associate with an object, such as a country or region. It is the result of a collection of organized associations within the cognitive system, rather than a simple objective description of the object, that incorporates past and future assumptions about its qualities and characteristics.

The cognitive aspect of the image is typically emphasized, even though the construct may include loosely related ideas and recollections. Although there may be ambiguities and contrasts, the image is typically coherent. Additionally, images contain emotive elements that can elicit positive or negative feelings about a country or region, based on political or social aspects. People's sense of community is part of their social structure and can impact their perceptions of a country or region, with shared objectives, social conventions, and values manifesting in the normative component of the country's image.

The term "image" can be divided into two categories: live-in image and made-in image. The former pertains to the country's sociocultural components, including its history and transactions, economic standing, social culture, values and norms, and political system. The latter refers to a country as a source of economic goods and services and captures consumer perceptions of the country of origin with respect to specific products. The qualities of the made-in image include those of the nation's product, knowledge, and image-shaping sectors and businesses [29].

## 2.3 Consumer buying behavior

The importance of consumer behavior in marketing is increasing in today's rapidly changing business world, as noted by Kaplan and Haenlein [30], Sangroya and Nayak [31], and Singh and Islam [32]. Marketing is a critical component of any organization or business, as it promotes business awareness and improves customer relationships [33]. Weak marketing strategies may hinder a company's ability to achieve its business goals.

Customers play a central role in developing marketing strategies, as they drive revenue for companies through their purchases and influence others to do the same, according to Khaniwale [34]. Understanding consumer buying behavior involves comprehending the decision-making process and the steps involved in purchasing products and services [35]. Consumer behavior encompasses a wide range of activities aimed at fulfilling the needs and desires of consumers, including problem recognition, idea formulation, and post-purchase behavior [35].

To develop a marketing strategy, the first step is to assess the market and consumer opportunities for a product or service, as suggested by Kotler [36]. Many researchers and marketers are studying consumer buying behavior, particularly in developing nations such as Nigeria [31, 37–39].

In the marketing context, consumers face challenges in reaching a decision due to the abundance of options available to them [35]. Consumer behavior refers to the study of people's motivations, thoughts, and needs in choosing one product over another, as well as their patterns in purchasing different products and services [35]. Several factors play a crucial role in the consumer decision-making process, and it is essential for the marketing team to have a clear understanding of these factors to effectively integrate them into their marketing strategies and reach consumers in a more impactful manner [34, 40, 41].

## 2.4 Steps in consumer buying behavior top of form

Marketers need to understand the consumer buying decision process, which refers to the series of steps taken by consumers in deciding to buy goods or services, as highlighted by Kotler [39]. To successfully sell products or services, marketers must comprehend the way consumers reach a purchasing decision. The consumer decision-making process involves five steps, which include need recognition, information search, evaluation of alternatives, purchase decision, and post-purchase behavior, as illustrated in Fig 1. Although some consumers may skip some steps, the final decision is influenced by their mindset and nature [39]. For instance, a consumer purchasing a regular brand of milk may skip information search and evaluation, while a consumer buying a high-involvement product may not. The decision-making process is more prominent for new purchases or high-involvement products, such as cars. While some companies focus on understanding the overall experience of the consumer in learning, choosing, using, and disposing of a product [42], this study will only concentrate on the steps taken in selecting a product or service.

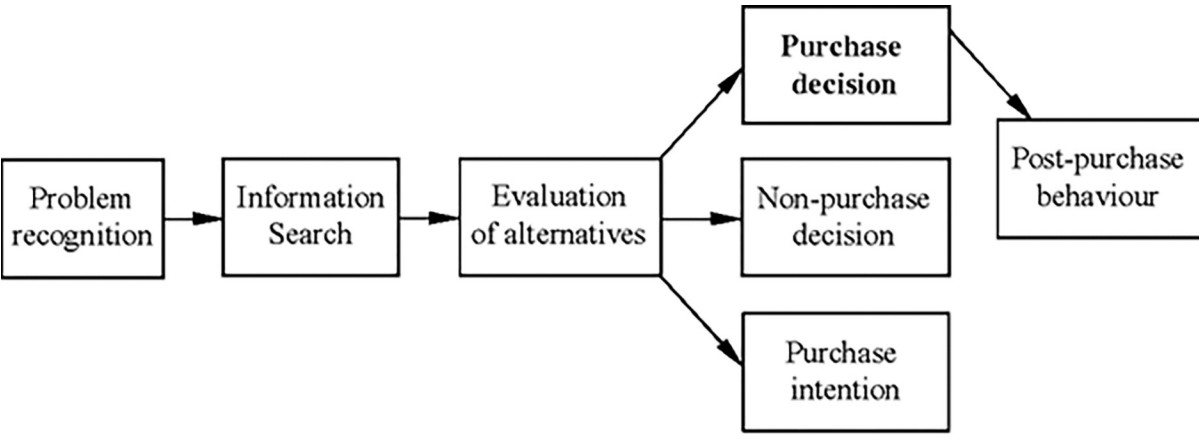

**Fig 1. Steps in consumer buying behaviour.** Source: Kotler & Keller [42].

**2.4.1 Need recognition.** According to Kotler and Keller [42] and Kotler [39], the initial stage in the consumer's purchasing decision-making process involves recognizing needs or problems, which could be a fundamental requirement for necessities such as food, drink, shelter, or air, or an extension of these necessities. Shma [43] suggests that at this stage, marketers must be attentive to customers' wants and focus on fulfilling them. To meet customer demands, marketers need to determine their needs and create marketing strategies tailored to meet those needs, as recommended by Kotler and Keller [42] and Kotler [39]. For example, although eating fulfills a basic need, a person's requirement may also include particular cuisine such as Nigerian jellof rice.

**2.4.2 Information search.** The second stage of the customer decision-making process involves customers recalling information about a product or service they are interested in purchasing. If a customer has had a positive experience with the product/service or knows someone who has, they may not feel the need to seek out further information because they trust the product/service. However, if a customer has had a negative experience or is considering a new product, they may seek out additional information [44, 45]. Customers use various sources to research products, including commercial sources (e.g., salespeople, commercials, social media, peer evaluations), personal sources (e.g., family, friends, acquaintances), and experimental sources (e.g., testing and using the product) [39]. For example, when someone wants to purchase a smartphone, they may pay attention to commercials and seek information from friends and family who have used the same smartphone model [45].

**2.4.3 Evaluation of alternatives.** Once the customer has gathered information about the product or service they are interested in, they move on to the third stage of the decision-making process, which involves evaluating potential alternatives. For instance, if a particular car brand does not meet the customer's preferences, requirements, tastes, or budget, they may consider researching other car brands that are available [42]. Because marketers may not have full insight into the customer's evaluation process or the factors that may impact their decision to choose one product over another, it can be challenging to comprehend this stage fully.

**2.4.4 Purchase decision.** According to Kotler and Keller [42], the phase in which the consumer decides to purchase a certain brand after considering available options involves gathering and assessing data from various sources, and choosing where or what to buy. During the assessment stage, the consumer typically chooses the product or brand that receives the highest rating. However, it should be noted that external factors may also influence the consumer's decision-making process.

**2.4.5 Post-purchase decision.** According to Kotler [39], marketers are aware that their responsibility goes beyond the moment of sale because the level of customer satisfaction or dissatisfaction after product usage can significantly impact their intention to repurchase and recommend the product to others. A satisfied customer is more likely to repurchase the product and even promote it to others, which can result in long-term brand loyalty and sustainable business performance. Post-purchase decision represents the intention of the customer to continue using the product or service.

## 2.5 Country of origin image and consumers' buying behavior

Numerous studies have examined the relationship between country of origin and consumer buying behavior. Fauser and Agola [6] conducted a study that explored how regional Italian images influence consumer behavior in Germany. They used data from 388 respondents, which were analyzed through multiple linear regression and paired t-tests. The results indicated that significant image differences exist and that these differences affect purchase probability. Moreover, the study found that country knowledge negatively impacts the predictive value of the measured regional image for purchase probability.

Jumani and Sukhabot [46] conducted a quantitative analysis using SMART-PLS to understand the importance of Islamic brand attitude and its impact on buying behavior among Malaysian Muslims. Their study aimed to uncover the role of Islamic branding in motivating Malaysian consumers to buy imported Islamic products. The results showed that country of origin was the most important aspect of an Islamic brand for Malaysian Muslims, followed by customer segment and compliance. This suggests that country of origin has a significant positive impact on buying behavior among Malaysian Muslims when it comes to Islamic brands.

Thøgersen and Pedersen [47] investigated the influence of an export country's environmental image on consumers' reactions to imported environmentally friendly products. They conducted an online survey in Denmark, Germany, and France, with 500 respondents from each country. The study found that consumers differentiated between a country's environmental image and its general and product-related images. Additionally, the environmental image of a country had a strong influence on consumers' evaluation of environmentally friendly products from that country.

In Thøgersen et al. [48], the authors examined the impact of a country's image on consumers' evaluations of imported products. They tested a hierarchical model in four countries (Germany, France, China, and Thailand) using a quantitative study of 1000 consumers from each country. The study evaluated the level of country image for Denmark and consumers' attitudes towards buying organic food products from Denmark. The data was analyzed using confirmatory factor analysis and structural equation modeling, and the results showed that the hierarchical country image model was a good fit for data in two European countries but not in two Asian countries. This suggests that hierarchical relationships as proposed by the model may require a high level of familiarity with the country and product type.

Akbarov [49] conducted a study to investigate the impact of consumer ethnocentrism on purchasing behavior and how this relationship is mediated by demographic factors. The study used questionnaires from 467 participants and found that the effects of ethnocentrism on buying behavior vary by product category and are mediated by marital status, gender, and personal income.

Dekhili et al. [50] examined the facets of a country's ecological image and the importance of geographic origin in sustainability. Their exploratory study used semi-structured interviews and focus groups and found that a country's ecological image has eight dimensions and that geographic origin is relevant for achieving sustainability.

Mahmoud et al. [51] investigated consumer xenocentrism and the intention to purchase foreign goods in an emerging economy, along with the role of cultural openness in this relationship. Their study used 204 responses collected through web-based sampling and found that except for country of origin and interpersonal influence, all other constructs positively and significantly influenced the intention to purchase foreign products. The study also found that country of origin, self-confidence, and self-esteem had an impact on the intention to purchase foreign products, but exposure to other cultures did not necessarily increase consumers' interest in foreign market products.

Fazli-Salehi et al. [52] aimed to examine the applicability of country affinity to domestic brands and the effect of nation sentiment on the self-brand connection of consumers with domestic versus foreign brands. Their study used an online survey in the United States and found that for foreign brands, consumers' self-brand connection was enhanced by country affinity and product quality evaluation. However, for domestic brands, the self-brand connection of consumers was influenced by their ethnocentric features, not country affinity or product quality evaluation.

Šapić et al. [10] investigated the influence of country of origin image on customer loyalty towards products from countries with a positive and recognizable image and evaluated if the effect was achieved through product features such as quality, design, and attractiveness. Their

cross-sectional quantitative study used 150 responses and found that the image of the country of origin is crucial to consumers and affects their decision to buy foreign products. The study also found that country of origin images influence consumer behavior when choosing products based on quality, design, and attractiveness, and have a significant impact on customer loyalty towards a brand.

Alonso Dos Santos et al. [53] explored whether a family member should communicate their family identity and country of origin in a cross-cultural study conducted in Chile and Spain. Their quantitative online survey found that communication of the family firm's identity increases brand trust and purchase intention, and consumers scored higher on trust and purchase intention when exposed to the national country of origin of a product.

Brucaj [54] conducted a study that examined the impact of country of origin and consumer ethnocentrism on Albanian consumers' perceptions of domestically produced goods. Through a cross-sectional quantitative study of 278 responses, the researcher found that ethnocentrism plays a crucial role in the evaluation of products by Albanian consumers and has a significant impact on their purchase intention. Additionally, the study revealed that country of origin also has a significant impact on the evaluation of foreign products in the Albanian market context.

In a quantitative survey of 293 French customers on four pasta brands, Bernard et al. [55] investigated the effect of the "made in the domestic country" (MIDC) strategy, which involves signaling a product's domestic origin to target domestic customers. The researchers discovered that the MIDC label increased customers' purchase intention but not their willingness to pay. The study also revealed that country of origin had a greater effect on consumer behavior when the product had less brand equity and consumers exhibited high ethnocentrism or were strongly attached to their national identity.

Potluri and Johnson [56] conducted research to determine the influence of country of origin on the buying decisions of consumers in the UAE. Through a survey of 370 respondents, they found that country of origin influences buying decisions for both products and services, but demographic factors did not have any impact.

Merabet [57] explored the impact of country of origin image on purchase intention and the variables of perceived quality and perceived price. The study of 120 respondents showed that country of origin image had a positive effect on perceived quality and perceived price, and perceived price mediated the relationship between country of origin and purchase intention.

Yunus and Rashid [58] investigated the factors of country of origin that Malaysian consumers consider when purchasing mobile phone brands from China. In a cross-sectional quantitative study of 200 responses, they found that all pre-determined variables, including country of origin, were significant and highly correlated in influencing consumer purchase intention in the mobile phone sector.

## 2.6 Mediating role of perceived quality

Numerous studies have examined the impact of price and quality on consumer buying behavior. Yu et al. [59] conducted a quantitative study to investigate the influence of brand origin, decision focus, and product quality on consumers' preferences. Their findings revealed that consumers from developing economies generally prefer foreign brands, particularly when purchasing for others and when the product quality is low. In a study by Cavite et al. [60], the authors explored the influence of product traceability knowledge, health consciousness, and subjective norms on the intention of consumers to purchase organic rice in Thailand. It was established that product quality plays a mediating role in this relationship.

Brandão and Costa [61] conducted a quantitative study to investigate the effects of barriers towards sustainable fashion consumption, extending the theory of planned behavior. Their

findings suggest that product attributes, specifically quality and variety, have mediating effects on the relationship between the theory of planned behavior and consumer purchase intentions. Xu et al. [62] developed a dual-systems model for online impulse buying and found that review quality, source credibility, and observational learning influence the perceived usefulness of online reviews, which in turn affects purchase decisions and impulse purchases. The perceived quality of a service also has an impact on consumer purchase intentions.

In a study conducted by Jo et al. [63] in the French fashion industry, researchers investigated the characteristics of multichannel shoppers and whether they prioritize quality or price. Findings showed that quality played a more significant role in the purchase decision of multichannel shoppers, and shoppers with high basket flexibility had a higher probability of becoming multichannel shoppers, with this probability increasing when the shopper is quality-oriented. Chi and Chen [64] conducted an online survey on Chinese consumers and found that perceived merchandise quality value, price value, emotional value, and aesthetic value of lifestyle fashion stores influence repurchase intention and time spent by consumers.

Garcia et al. [65] studied online group buying and found that service quality, popularity, and online brand image positively affect consumers' overall satisfaction, purchase intentions, and loyalty, with quality being a strong determinant of loyalty. Santy and Atika [66] examined the effect of quality perception and product knowledge on the purchase decision of customers in the Indonesian smartphone market and found that quality perception has a significant and positive influence on customers' purchase decision when buying foreign-manufactured smartphones in Indonesia. Top of Form.

Faisal-E-Alam [67] conducted a study in Bangladesh which compared local and multinational cosmetics firms to investigate the impact of quality on consumers' purchase intentions. The findings of the study, based on 1,167 samples, suggest that quality plays a critical role in the decisions of customers, especially when selecting cosmetics brands from multinational firms. Furthermore, the study also found that customers tend to avoid purchasing local brands due to quality-related issues, which indicates that the country of origin significantly affects customers' purchase decisions.

Khare [68] investigated the influence of past environmental behavior, green peer influence, and green apparel knowledge on Indian consumers' perceived benefits of green apparel. The study was conducted using the mail intercept technique across 10 cities in India. According to the findings, previous environmental behavior, green peer influence, and green apparel knowledge all have an impact on the perceived benefits of green apparel. The perceived benefits included attributes like awareness of fair trade practices, perceived value in buying fair trade clothes, and the enhancement of image. The study also found that quality plays a role in these relationships.

Bukhari et al. [69] conducted a phenomenological study on the motives behind the purchase of western-imported food products from a Muslim-dominated region. The study was quantitative, and in-depth, semi-structured interviews were conducted with 90 participants across Pakistan. The findings suggest that the product attributes of quality, including design size, attractive packaging colors, overall quality of materials used, labeling with maximum product information, and taste, significantly influence consumers' purchasing behavior. The study also revealed that the vast majority of respondents preferred products that are halal, which was a crucial determinant in their purchase decision.

Tariq et al. [70] aimed to examine the impact of consumers' attitudes towards organic foods on online impulse buying behavior, and the mediating effect of three website features (informational, visual, and navigational design) on this relationship. Data were collected through an online survey of 653 respondents, and SEM was used for data analysis. The findings suggest that Chinese consumers' attitudes towards organic products are influenced by social media

forums, ratings, and reviews, which positively influence their online impulse buying behavior. However, this influence is mediated by product quality, webpage features, and certifications.

Sinaga and Evi [71] investigated perception analysis and consumer behavior in purchasing Shell Helix lubricant oil products manufactured by PT. Tira Wira Usaha. Their study suggests that customers' perceptions of quality toward product attributes significantly influence their purchasing behavior.

Al-Huwaishel and Meshal [72] conducted a study focusing on Saudi females to examine the impact of perceived value, quality, and loyalty on purchase intention in the accessories sector. Based on the analysis of 170 responses, the study found that quality and value have a significant positive impact on customers' purchase decisions, as well as their repurchase and recommendation intentions. In other words, the higher the quality, the higher the likelihood of customers purchasing a given cosmetics brand.

## 2.7 Hypothesis

**H1:** *Country of origin image is not significantly important in consumers' search for information when buying from Nigerian hypermarkets.*

**H2:** *Country of origin image does not significantly influence consumers' evaluation of alternative brands when buying from Nigerian hypermarkets.*

**H3:** *Country of origin image does not significantly influence consumers' purchase decision when buying from Nigerian hypermarkets.*

**H4:** *Country of origin image is not significantly important in consumers' product loyalty when buying from Nigerian hypermarkets.*

**H5:** *The relationship between country of origin image and consumers' buying behaviour in Nigerian hypermarkets is not mediated by perceived quality of product.*

## 3. Research method

This study employed a cross-sectional research design and was conducted in Nigeria through an online survey that targeted responses from all over the country. To mitigate bias, an IP blocker was used on Google Forms to only allow Nigerian IP addresses to participate, and the survey was promoted through various social media and online platforms. The survey utilized a structured questionnaire, which was divided into two sections. The first section used Likert's rating scale to gather data about variables considered in the study, while the second section collected dichotomous data about consumers' demographics.

To ensure conformity with existing standards and content validity, the appropriate sample size was determined based on past related empirical works, as recommended by Cochran [73] and this is documented in Table 1. A total of 1272 responses were gathered, and the data was

**Table 1. Battery of measurement items and sources.**

| Latent construct | Number of items | Source(s) of scale |
|---|---|---|
| Country of Origin Image | 8 | Fazli-Salehi et al. [52] and Šapić et al. [10] |
| Consumer Behaviour: | | |
| • Information search | 3 | Dekhili et al. [50] and Mahmoud et al. [51] |
| • Evaluation of alternatives | 3 | |
| • Purchase decision | 3 | |
| • Loyalty (post-purchase behaviour) | 3 | |
| Mediating Role of Perceived Quality | 5 | Brandão and Costa [61] |

Source: Authors (2023)

**Table 2. Distribution and return of questionnaire.**

| Category of respondents | No of copies of questionnaire distributed | No of usable copies of questionnaires returned | Number of loss and invalid copies of questionnaires | % returned | % lost |
|---|---|---|---|---|---|
| Customer of hypermarkets in Nigeria | Unlimited (Online survey) | 1271 | 0 | 100 | 0 |

analyzed using AMOS-SEM software. The study's results are documented in the results section, and the original data can be accessed via Kaggle (https://doi.org/10.34740/KAGGLE/DSV/4972216)

This study had a total of 1272 valid responses, as seen in Table 2. This is because the online survey used in the study had a feature that prevented the submission of incomplete questionnaires, requiring participants to answer all questions before submitting their responses. As a result, the study achieved a 100% valid response rate. This response rate is similar to the findings of Thøgersen et al. [48], who obtained 4,000 responses from four countries, with 1,000 responses from each country.

## 3.1 Inclusivity in global research–Ethical statement

Since the first author wasn't a Nigerian, there was a need for an ethical committee to approve the entire research process under the guidelines contained in the inclusivity of global research. This approval was issued by the Marketing Department, under the Faculty of Management Sciences, at the Enugu State University of Science and Technology. The approval was signed by the Head of Department, Professor Ikenna Chukwu. The research was approved because the first author had worked on other studies with the second and fourth authors, with the fourth author being a member of the approving department, and the need for their continued collaboration in the academic sector was considered pivotal in reaching the decision. Additionally, the first author was more experienced, and it was deemed necessary that the second author work under his guidance. Finally, the department set up a committee that monitored the entire research process to ensure that it was in line with established ethical standards in the academic world but did not sponsor the study. The committee was headed by Professor Gerald Nebo, with Dr. Jude Eze as secretary.

Respondents' written consent was sought. To participate in the study, the respondents had to agree to have their data used for analysis by signing the first page, which contained the consent and privacy statement.

## 3.2 Data analysis

**3.2.1 Demographic variables.** Table 3 presents the results of the demographic analysis, indicating that the majority of the respondents were male (64.2%), with female participants accounting for 34.8%. The largest age group represented was 31–40 years (44%), followed by those over 50 years old (21.1%), 20–30 years (19.8%), 41–50 years (12.6%), and the youngest group, below 20 years, making up only 2.5% of the respondents. Ethnicity-wise, the majority of respondents were Igbo (49%), followed by Middle Belt (22.2%), Yoruba (13.5%), Others (9%), and Hausa/Fulani (6.3%).

To ensure that the responses were informed, participants were asked if they had prior shopping experience in a hypermarket, with 95.6% responding in the affirmative, indicating that the majority of respondents were suitable for participating in this study as they have shopped in a hypermarket before. The frequency of their visits to a hypermarket was also assessed, with the largest group (35.2%) stating that they visit "at least once per month", followed by "seldom"

**Table 3. Demographic variables.**

| | | Frequency | Percent % | Cumulative percent % |
|---|---|---|---|---|
| Gender | Male | 816 | 64.2 | 64.2 |
| | Female | 456 | 35.8 | 100 |
| Age | Below 20 years old | 32 | 2.5 | 2.5 |
| | 20–30 years old | 252 | 19.8 | 22.3 |
| | 31–40 years old | 560 | 44.0 | 66.4 |
| | 41–50 years old | 160 | 12.6 | 78.9 |
| | Above 50 years old | 268 | 21.1 | 100.0 |
| Ethnicity | Hausa/Fulani | 80 | .6.3 | .6.3 |
| | Igbo | 624 | 49.0 | 55.3 |
| | Yoruba | 172 | 13.5 | 68.8 |
| | Middle-Belt | 282 | 22.2 | 91 |
| | Others | 114 | 9.0 | 100.0 |
| Have you ever shopped in a hypermarket before? | Yes | 1216 | 95.6 | 95.6 |
| | No | 56 | 4.4 | 100.0 |
| How often do you shop in hypermarkets? | Never | 44 | 3.5 | 3.5 |
| | Seldom | 348 | 27.4 | 30.8 |
| | At least once per year | 188 | 14.8 | 45.6 |
| | At least once per month | 448 | 35.2 | 80.8 |
| | At least once per week | 244 | 19.2 | 100.0 |
| Do you prefer buying products from a certain country (like Italian Wines), because the country is known for manufacturing such product? | Yes | 796 | 62.6 | 62.6 |
| | No | 476 | 37.4 | 100.0 |

(27.4%), "at least once per week" (19.2%), "at least once per year" (14.8%), and "never" (3.5%). Lastly, respondents were asked if they had a preference for buying products from a specific country, with 62.6% saying "yes" and 37.4% saying "no".

Overall, the analysis suggests that the responses gathered are representative of Nigeria, regardless of ethnicity or geography, and the results can be applied to the entire country.

### 3.3 Assessment of measurement model

As explained in the methodology section, before conducting the structural model assessment, the researcher assessed the reliability and validity of the data. The term "reliability" refers to the consistency with which a scale measures a particular construct [74]. To evaluate the convergent validity of the data, the study employed various tests such as Composite Reliability (CR), Cronbach's Alpha, Squared Shared Variance (ASV), factor loadings, Maximum Shared Variance (MSV), and Average Variance Extracted (AVE), as documented in Table 4. The results indicated that the Composite reliability and Cronbach's alpha values were higher than 0.7, suggesting good reliability of all the constructs employed in the study [75]. Moreover, the AVE value was greater than 0.5, indicating the convergent validity of the data [76, 77]. According to Wang and Shah [78], for the convergent validity to be deemed valid, the values for both ASV and MSV should be less than those of the AVE. Table 4 illustrates that all ASV and MSV values are lower than the AVE, thereby confirming the validity of the convergent validity [78].

The conventional approach for assessing discriminant validity, proposed by Fornell and Larcker [79], was employed by the researcher. According to this approach, discriminant validity is established when the square roots of the Average Variance Extracted (AVE) for each

**Table 4. Convergent validity.**

|  | Factor loading | Crombach α | CR | AVE | ASV | MSV |
|---|---|---|---|---|---|---|
| COI8 | 0.744 | 0.93 | 0.91 | 0.58 | 0.41 | 0.50 |
| COI7 | 0.768 | | | | | |
| COI6 | 0.758 | | | | | |
| COI5 | 0.701 | | | | | |
| COI4 | 0.808 | | | | | |
| COI3 | 0.803 | | | | | |
| COI2 | 0.717 | | | | | |
| COI1 | 0.788 | | | | | |
| IS1 | 0.835 | 0.92 | 0.88 | 0.71 | 0.66 | 0.53 |
| IS2 | 0.817 | | | | | |
| IS3 | 0.884 | | | | | |
| EOA1 | 0.887 | 0.91 | 0.91 | 0.77 | 0.73 | 0.64 |
| EOA2 | 0.957 | | | | | |
| EOA3 | 0.79 | | | | | |
| PD1 | 0.643 | 0.84 | 0.756 | 0.51 | 0.47 | 0.46 |
| PD2 | 0.692 | | | | | |
| PD3 | 0.799 | | | | | |
| PP1 | 0.804 | 0.81 | 0.76 | 0.51 | 0.50 | 0.43 |
| PP2 | 0.675 | | | | | |
| PP3 | 0.675 | | | | | |
| MRQ5 | 0.865 | 0.93 | 0.90 | 0.65 | 0.51 | 0.48 |
| MRQ4 | 0.707 | | | | | |
| MRQ3 | 0.83 | | | | | |
| MRQ2 | 0.755 | | | | | |
| MRQ1 | 0.867 | | | | | |

construct are greater than the correlation values between other constructs. In addition, cross-loading was also used. Table 5 shows that the first value in each column is greater than the other values in the same column, indicating that discriminant validity has been established for the loaded data [79].

The study evaluated the model's appropriateness using three measures: TLI, CFI, and RMSEA. For a good fit, TLI and CFI should exceed 0.90, which was observed in this study. Additionally, a good fit for RMSEA is indicated by a value less than 0.10, which was also observed in Table 6. Therefore, based on the results presented in the table, it can be concluded that the model has a good fit.

Based on the discussions mentioned earlier, the data gathered in this study is suitable for further analysis. This is because the data has both reliable convergent and discriminant

**Table 5. Discriminant validity—Fornell and Larcker criterion results.**

|  | COI | IS | EOA | PD | PPD | MRQ |
|---|---|---|---|---|---|---|
| COI | **0.761577** | | | | | |
| IS | 0.745 | **0.842615** | | | | |
| EOA | 0.617 | 0.551 | **0.877496** | | | |
| PD | 0.58 | 0.801 | 0.552 | **0.714143** | | |
| PPD | 0.544 | 0.868 | 0.648 | 0.865 | **0.714143** | |
| MRQ | 0.616 | 0.469 | 0.369 | 0.603 | 0.413 | **0.806226** |

**Table 6. Model goodness of fit.**

| Selected indices | Result | Acceptable level of fit |
|---|---|---|
| TLI | 0.903 | TLI > 0.90 |
| CFI | 0.901 | CFI > 0.90 |
| RMSEA | 0.003 | RMSEA < 0.05 good; 0.05 to 0.10 acceptable |

validity, and the model also fits well. Therefore, additional analysis related to the hypothesis proposed can be presented and discussed in the following sections.

### 3.4 Test of hypothesis

**3.4.1 H1-H4—Country of origin image and consumer buying behavior.** The results of the latent variable path analysis are displayed in Fig 2 and Table 7. Fig 2 demonstrates that the country of origin has a positive association with four other factors: information search (0.83), evaluation of alternatives (0.56), purchase decision (0.66), and post-purchase decision (0.81). These correlations are highly significant, as shown by the beta values and statistical significance in Table 7, which is less than 0.001. Consequently, null hypotheses 1–4 are rejected.

**3.4.2 Hypothesis 5 –Mediating role of perceived quality.** Fig 3 and Table 8 depict the path analysis conducted for hypothesis (H5), which examines the effect of perceived quality on

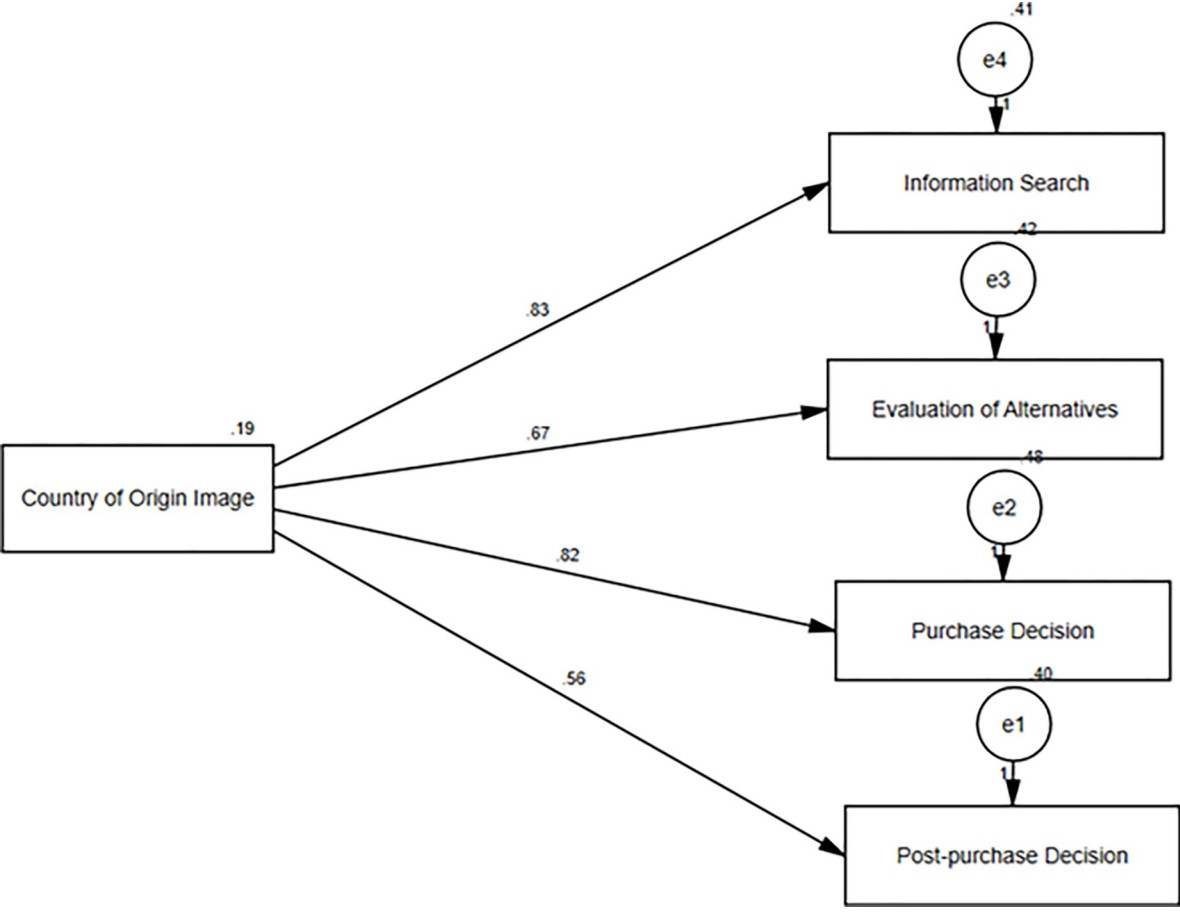

**Fig 2. Path analysis.**

**Table 7. Path analysis.**

| Relationships | | | Estimate | Beta | S.E. | C.R. | P |
|---|---|---|---|---|---|---|---|
| IS | <— | COI | .831 | .495 | .041 | 20.322 | 0.000 |
| PPD | <— | COI | .564 | .366 | .040 | 14.019 | 0.000 |
| EOA | <— | COI | .668 | .412 | .041 | 16.139 | 0.000 |
| PD | <— | COI | .816 | .459 | .044 | 18.425 | 0.000 |

the relationship between country of origin image and consumers' purchasing behavior. As indicated in Table 8, the direct effect of the country of origin image on consumers' purchasing behavior is significant (p<0.05) for all factors measured. However, the indirect effect, mediated by perceived quality, is significant (p<0.05) only for purchase decisions and post-purchase decisions and not significant for information searches and evaluation of alternatives.

Table 9 presents information about how certain factors influence purchase decisions and post-purchase decisions. The table shows three types of effects: total, direct, and indirect. However, only the effects related to purchase decisions and post-purchase decisions are significant. Therefore, the discussion will focus mainly on those two effects. The table shows that there is partial mediation, meaning that the effect is not the only factor influencing the decisions. The effect is strongest for post-purchase decisions. Overall, the hypothesis (H5) is partly supported in its original form.

## 4. Discussion and conclusion

The examination began with a focus on data visualization, which indicated that the online questionnaires had a 100% response rate. This was achieved by setting up the questionnaire in

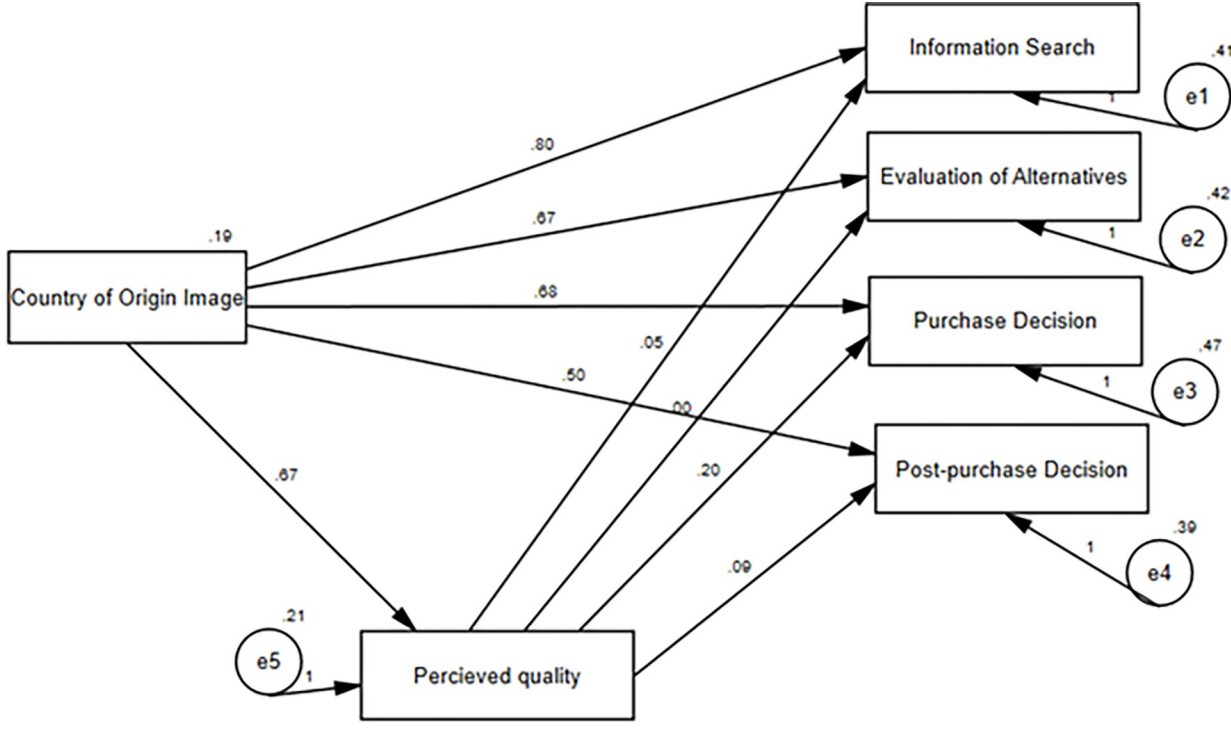

**Fig 3. Path analysis for H5.**

**Table 8. Path analysis for H5.**

| Relationship | | | Estimate | Beta | S.E. | C.R. | P |
|---|---|---|---|---|---|---|---|
| MRQ | <— | COI | .666 | .541 | .029 | 22.956 | 0.000 |
| PPD | <— | COI | .504 | .327 | .048 | 10.541 | 0.000 |
| PD | <— | COI | .681 | .383 | .052 | 13.053 | 0.000 |
| EOA | <— | COI | .670 | .413 | .049 | 13.595 | 0.000 |
| IS | <— | COI | .796 | .474 | .049 | 16.373 | 0.000 |
| PPD | <— | MRQ | .091 | .073 | .039 | 2.348 | .019 |
| PD | <— | MRQ | .202 | .140 | .042 | 4.759 | 0.000 |
| EOA | <— | MRQ | -.002 | -.001 | .040 | -.047 | .963 |
| IS | <— | MRQ | .053 | .039 | .040 | 1.340 | .180 |

a way that necessitated respondents to answer all questions before submitting it. As a result, all questionnaires submitted were completed, resulting in a 100% response rate.

The demographic analysis revealed that the majority of respondents were male (64.2%), aged between 31–40 years old (44%), of Igbo ethnicity (49%), had shopped in a Nigerian hypermarket before (95.6%), shop at a Nigerian hypermarket at least once per month (35.2%), and preferred purchasing products from a particular country (62.6%). Thus, the study's research questions were well addressed since the respondents had characteristics that aligned with the research objectives.

Prior to conducting data analysis, convergent reliability and discriminant validity tests were performed. The study found that the Crombach's alpha and composite reliability values for all constructs were above 0.7, indicating that the selected constructs were reliable. The AVE values for all constructs were above the minimum threshold of 0.50, and the ASV and MSV values were lower than the AVE, confirming the convergent validity. Discriminant validity was established by comparing the correlation value between constructs to the square root of their AVEs. Furthermore, the goodness of fit analysis results indicated that the TLI and CFI values were above 0.90, and the RMSEA value was below 0.10, suggesting a good model fit for further analysis.

After establishing the reliability, validity, and goodness of fit, the next step was to test the hypotheses. The study examined the direct relationship between the independent and dependent variables to test hypotheses (H1-H4). Results indicated that the country of origin image positively influenced information search (0.83), evaluation of alternatives (0.56), purchase decision (0.66), and post-purchase decision (0.81). This implied that hypotheses 1–4 were rejected in their null forms, and country of origin image was found to influence the consumers' decision-making process while shopping in Nigerian hypermarkets. This aligns with previous studies that have reported that country of origin image influences consumers' purchase

**Table 9. Total effect, direct effect and indirect effect.**

| | Total effect | | Direct effect | | Indirect effect | |
|---|---|---|---|---|---|---|
| | COI | MRQ | COI | MRQ | COI | MRQ |
| MRQ | .541 | .000 | .541 | .000 | .000 | .000 |
| IS | .495 | .039 | .474 | .039 | .021 | .000 |
| EOA | .412 | -.001 | .413 | -.001 | -.001 | .000 |
| PD | .459 | .056 | .383 | .140 | .076 | .040 |
| PPD | .366 | .088 | .327 | .073 | .039 | .120 |

decisions [6, 10, 46–52]. The study concluded that the country of origin image significantly and positively affects the buying behavior of consumers in Nigerian hypermarkets.

In this study, the researchers also examined the relationship between country of origin image, perceived quality, and consumer purchasing behavior. The results showed that perceived quality mediates the relationship between country of origin image and consumer purchasing behavior, but only partially, specifically for purchase decisions and post-purchase decisions. This finding is consistent with previous research [59–61], which has also found that perceived quality mediates the relationship between country of origin image and consumers' purchase behavior, although only partially.

Thus, the study confirms that country of origin image has a significant impact on consumers' purchasing behavior, including their information search, evaluation of alternatives, purchase decision, and post-purchase decision. However, the role of perceived quality is only partial, as it only affects the relationship between country of origin image and purchase decisions and post-purchase decisions.

## Author Contributions

**Conceptualization:** João Luis Lucas, Chiemelie Benneth Iloka.

**Data curation:** Shedrack Chinwuba Moguluwa, Chiemelie Benneth Iloka.

**Formal analysis:** Shedrack Chinwuba Moguluwa, Chiemelie Benneth Iloka, Mário Nuno Mata.

**Funding acquisition:** José Moleiro Martins.

**Investigation:** Shedrack Chinwuba Moguluwa.

**Methodology:** Shedrack Chinwuba Moguluwa, Mário Nuno Mata.

**Project administration:** Shedrack Chinwuba Moguluwa.

**Resources:** José Moleiro Martins, Mário Nuno Mata.

**Software:** José Moleiro Martins, Mário Nuno Mata.

**Supervision:** José Moleiro Martins, João Luis Lucas.

**Validation:** José Moleiro Martins, João Luis Lucas.

**Visualization:** José Moleiro Martins, João Luis Lucas.

**Writing – original draft:** João Luis Lucas, Chiemelie Benneth Iloka, Mário Nuno Mata.

**Writing – review & editing:** João Luis Lucas, Chiemelie Benneth Iloka, Mário Nuno Mata.

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
