## [Decision Letter · Decision Letter 0]

6 Apr 2023

PONE-D-23-04841Does Perceived Quality Mediate the Relationship between Country of Origin Image and Consumer Buying Behaviour in Nigerian Hypermarkets?PLOS ONE

Dear Dr. Moguluwa,

Thank you for submitting your manuscript to PLOS ONE. After careful consideration, we feel that it has merit but does not fully meet PLOS ONE’s publication criteria as it currently stands. Therefore, we invite you to submit a revised version of the manuscript that addresses the points raised during the review process.

ACADEMIC EDITOR: Minor Revision *Please provide a final paper with all revisions made and with an additional check on the level of plagiarism and compliance with the editorial Journal's guidelines.*==============================

We look forward to receiving your revised manuscript.

Kind regards,

Vincenzo Basile, PhD

Academic Editor

PLOS ONE

Journal Requirements:

Reviewers' comments:

Reviewer's Responses to Questions

**Comments to the Author**

1. Is the manuscript technically sound, and do the data support the conclusions?

Reviewer #1: Yes

Reviewer #2: Yes

2. Has the statistical analysis been performed appropriately and rigorously? 

Reviewer #1: Yes

Reviewer #2: Yes

3. Have the authors made all data underlying the findings in their manuscript fully available?

Reviewer #1: Yes

Reviewer #2: Yes

4. Is the manuscript presented in an intelligible fashion and written in standard English?

Reviewer #1: Yes

Reviewer #2: Yes

5. Review Comments to the Author

Reviewer #1: I enjoyed reading the piece of research.

It is very clearly organizes, written in good English.

I have just a few suggestions that could improve the paper.

I have not particularly enjoyed the chapter regarding consumer behavior. I think more interesting and newer information could have been presented regarding this aspect.

At the end of the article we should be able to see the limits of the research and the future research suggestions, and the implications for practitioners and researchers. The contributions of this paper should also be stated and underlined in the Conclusions chapter.

Reviewer #2: This research is good. However to make it better, please clarify the methodology parts and indicate the sampling methods and sampling process as they are ambiguous. Please revisit your references and properly write them alphabetically if possible unless it fulfilled the journal requirements. Please also re-consider the length of this writeup and be precise on your literature review. Your conclusion is not comprehensive enough to support the whole study and please also indicate the implication of this research practically and theoretically. If possible, please mention clearly the future research needs. Best wishes.

6. PLOS authors have the option to publish the peer review history of their article (what does this mean?). If published, this will include your full peer review and any attached files.

Reviewer #1: No

Reviewer #2: **Yes: **Arman Hj. Ahmad

---

## [Author Response · Author response to Decision Letter 0]

11 Apr 2023

Response is attached as a separate file "Response to Reviewers"

---

## [Editor Report · Decision Letter 1]

24 Apr 2023

Does Perceived Quality Mediate the Relationship between Country of Origin Image and Consumer Buying Behaviour in Nigerian Hypermarkets?

PONE-D-23-04841R1

Dear Dr. Shedrack Chinwuba Moguluwa,

We’re pleased to inform you that your manuscript has been judged scientifically suitable for publication and will be formally accepted for publication once it meets all outstanding technical requirements.

Kind regards,

Vincenzo Basile, PhD

Academic Editor

PLOS ONE

Additional Editor Comments:

Please provide a final paper with all revisions made and I recommend an additional check on plagiarism and/or compliance with the Journal's guidelines.
---

## [Editor Report · Acceptance letter]

9 May 2023

PONE-D-23-04841R1 

Does Perceived Quality Mediate the Relationship between Country of Origin Image and Consumer Buying Behaviour in Nigerian Hypermarkets? 

Dear Dr. Moguluwa:

I'm pleased to inform you that your manuscript has been deemed suitable for publication in PLOS ONE. Congratulations! Your manuscript is now with our production department. 

Kind regards, 

on behalf of

Dr. Vincenzo Basile 

Academic Editor

PLOS ONE